# Therapeutic Vaccine in Chronically HIV-1-Infected Patients: A Randomized, Double-Blind, Placebo-Controlled Phase IIa Trial with HTI-TriMix [note 1]

**DOI:** 10.3390/vaccines7040209

**Published:** 2019-12-06

**Authors:** Wesley de Jong, Lorna Leal, Jozefien Buyze, Pieter Pannus, Alberto Guardo, Maria Salgado, Beatriz Mothe, Jose Molto, Sara Moron-Lopez, Cristina Gálvez, Eric Florence, Guido Vanham, Eric van Gorp, Christian Brander, Sabine Allard, Kris Thielemans, Javier Martinez-Picado, Montserrat Plana, Felipe García, Rob A. Gruters

**Affiliations:** 1Department of Viroscience, Erasmus MC, 3015 Rotterdam, The Netherlands; w.dejong.2@erasmusmc.nl (W.d.J.); e.vangorp@erasmusmc.nl (E.v.G.); 2Infectious Diseases Department, Hospital Clínic-HIVACAT, University of Barcelona, 08036 Barcelona, Spain; LALEAL@clinic.cat; 3Institut d’Investigacions Biomèdiques August Pi i Sunyer (IDIBAPS)-HIVACAT, 08036 Barcelona, Spain; Alberto.Crespo@bd.com (A.G.); MPLANA@clinic.cat (M.P.); 4Clinical Trials Unit, Clinical Sciences Department, Institute of Tropical Medicine of Antwerp, 2000 Antwerp, Belgium; jbuyze@itg.be; 5Virology Unit, Biomedical Sciences Department, Institute of Tropical Medicine of Antwerp, 2000 Antwerp, Belgium; pieter.pannus@gmail.com (P.P.); eflorence@itg.be (E.F.); GVanham@itg.be (G.V.); 6IrsiCaixa AIDS Research Institute—HIVACAT, Hospital Germans Trias i Pujol, 08916 Badalona, Spain; msalgado@irsicaixa.es (M.S.); bmothe@irsicaixa.es (B.M.); saramoronlopez@gmail.com (S.M.-L.); cgalvez@irsicaixa.es (C.G.); cbrander@irsicaixa.es (C.B.); Jmpicado@irsicaixa.es (J.M.-P.); 7Fundació Lluita contra la Sida, Infectious Diseases Department, Hospital Germans Trias i Pujol, 08916 Badalona, Spain; jmolto@flsida.org; 8University of Vic—Central University of Catalonia (UVic-UCC), 085000 Vic, Spain; 9Department of Infectious Diseases, Erasmus MC, 3015 Rotterdam, The Netherlands; 10Catalan Institution for Research and Advanced Studies (ICREA), 08010 Barcelona, Spain; 11Department of Internal Medicine and Infectious Diseases, Universitair Ziekenhuis Brussel, 1090 Brussels, Belgium; Sabine.Allard@uzbrussel.be; 12eTheRNA, BVBA (eTheRNA), 2845 Niel, Belgium; Kris.Thielemans@vub.ac.be; 13Laboratory of Molecular and Cellular Therapy, Vrije Universiteit Brussel (VUB), 1090 Brussels, Belgium

**Keywords:** HIV-1, therapeutic vaccine, functional cure, immunotherapy, mRNA, lymph node, TriMix

## Abstract

Therapeutic vaccinations aim to re-educate human immunodeficiency virus (HIV)-1-specific immune responses to achieve durable control of HIV-1 replication in virally suppressed infected individuals after antiretroviral therapy (ART) is interrupted. In a double blinded, placebo-controlled phase IIa multicenter study, we investigated the safety and immunogenicity of intranodal administration of the HIVACAT T cell Immunogen (HTI)-TriMix vaccine. It consists of naked mRNA based on cytotoxic T lymphocyte (CTL) targets of subdominant and conserved HIV-1 regions (HTI), in combination with mRNAs encoding constitutively active TLR4, the ligand for CD40 and CD70 as adjuvants (TriMix). We recruited HIV-1-infected individuals under stable ART. Study-arms HTI-TriMix, TriMix or Water for Injection were assigned in an 8:3:3 ratio. Participants received three vaccinations at weeks 0, 2, and 4 in an inguinal lymph node. Two weeks after the last vaccination, immunogenicity was evaluated using ELISpot assay. ART was interrupted at week 6 to study the effect of the vaccine on viral rebound. The vaccine was considered safe and well tolerated. Eighteen percent (*n* = 37) of the AEs were considered definitely related to the study product (grade 1 or 2). Three SAEs occurred: two were unrelated to the study product, and one was possibly related to ART interruption (ATI). ELISpot assays to detect T cell responses using peptides covering the HTI sequence showed no significant differences in immunogenicity between groups. There were no significant differences in viral load rebound dynamics after ATI between groups. The vaccine was safe and well tolerated. We were not able to demonstrate immunogenic effects of the vaccine.

## 1. Introduction

Despite the major successes of antiretroviral treatment (ART), the burden of treatment and costs drive the search for alternatives [1,2]. Therefore, alternative strategies to achieve a durable control of human immunodeficiency virus (HIV) replication are needed. Therapeutic vaccinations, aiming at the induction or enhancement of HIV-specific immune responses, represent a promising approach [3]. In most therapeutic vaccine trials, the products tested were safe and induced CD4 and/or CD8 T cell-mediated immune responses [4,5]. However, this did not result in prolonged suppression of HIV replication in patients that underwent ART interruption [5]. For this reason, novel immunogens in combination with new adjuvants are being investigated.

Here, we explore the use of a new HIV immunogen, known as HIVACAT T cell Immunogen (HTI), in a phase IIa clinical trial, conducted by the iHIVARNA consortium (see: Appendix B). Its mRNA sequence joins critical HIV CTL epitopes that are overall conserved and predominantly recognized by individuals with spontaneous low HIV viral loads. CTL responses against the HTI immunogen had a higher functional avidity and broader cross-reactivity than CTL targeting other HIV antigens [6,7,8]. Eliciting or expanding this type of CTL responses might, therefore, induce T cell-mediated control of viral replication. As adjuvants, TriMix, a rationally designed product with mRNAs encoding a constitutively active form of Toll-like receptor 4 (caTLR4), the ligand for CD40 (CD40L) and CD70 was used, which has been shown to promote antigen presenting cells function [9].

In preclinical mouse experiments, HTI-TriMix was found to be immunogenic [10,11]. Next, HTI-TriMix was proved to be safe in a dose-escalating phase I clinical trial in 21 chronic HIV-1-infected patients under stable ART [10,12]. In the current randomized placebo-controlled double-blinded study [13], HIV-1-infected individuals on stable ART were vaccinated with HTI-TriMix, TriMix alone, as control, or Water for Injection (WFI), as placebo, to investigate safety and T cell immunogenicity of HTI-TriMix. In addition, after vaccination, ART was interrupted to study the effect of vaccination on viral rebound. Furthermore, we report the effect on viral DNA and RNA reservoir. After completion of the phase I and IIa trials, it became apparent that the study product HTI-TriMix contained an unintended start codon upstream of the HTI immunogen coding sequence. This error could have influenced the expression of the mRNA-encoded HTI vaccine protein (see: Appendix C).

## 2. Materials and Methods

### 2.1. Patients

Inclusion started after approval (see Appendix D for trial registration; reference numbers between parentheses) from designated ethical committees (EC) of the participating centers and competent authorities of The Netherlands (NL57593.000.16), Belgium (2016-002724-83), and Spain (2016-002724-83). From March until July 2017, we recruited HIV-1-infected participants under stable ART from participating centers Erasmus MC (MEC-2016-602, Rotterdam, The Netherlands), Hospital Clinic, (25/2016, Barcelona, Spain), Hospital Germans Trias i Pujol, (EC: Hospital Clinic, Badalona, Spain), Institute of Tropical Medicine (EC: Universitair Ziekenhuis Brussel, Antwerp, Belgium), and Universitair Ziekenhuis Brussel (2016/316- Brussels, Belgium). Comprehensive inclusion and exclusion criteria are available in Appendix A. Briefly, patients were eligible for enrollment if stably treated HIV-1 infection, as determined by a viral load (pVL) ≤ 50 copies/mL, a CD4 cell count ≥ 450 cells/μL, and a nadir CD4 T cell count ≥ 350 cells/μL. Female participants were closely monitored for possible pregnancy. During ART interruption (ATI), reinforced counselling for HIV transmission prevention was provided. Written informed consent was obtained, prior to inclusion.

### 2.2. Vaccinations

After screening, eligible patients were enrolled and randomized to one of the three study-arms, as seen in Figure 1A—layout according to the Consolidated Standards of Reporting Trials (CONSORT) statement [14]. Per protocol, each study arm (HTI-TriMix *n* = 40, TriMix *n* = 15 or WFI *n* = 15) was randomly allocated and stratified per center. This ratio would allow sufficient study of HTI-TriMix compared to control groups TriMix and WFI, as is detailed below and in Appendix A. An interim analysis was prespecified once half of the patients were recruited. No patients were further recruited until the results of this interim analysis were analyzed (see below). Randomization and blinding were performed by an independent statistician, using R for the generation of codes [15]. The vaccine was administered using ultrasound guidance, by a trained radiologist in an inguinal lymph node (visits at week 0, 2, and 4). ATI started two weeks after last vaccination, at week 6 and during up to 12 weeks to week 18, with in-hospital follow-up, including physical examination, as well as viral load and CD4 T cell count measurements at weeks 8, 10, and 14, as seen in Figure 1B. ART could be reinitiated at any time before week 18 if CD4 T cell counts dropped below 50% of baseline or to <350 cells/μL, and/or if clinically recommended. After ART was resumed, patients were followed until 12 weeks to assess pVL and CD4 T cell count recovery, as seen in Figure 1B.

### 2.3. Study Safety (Primary Objective)

Safety assessments consisted of subject diary cards up to seven days following the vaccinations, outpatient clinic visits, and routine laboratory assessments, as seen in Figure 1B. Safety assessments were categorized according to Common Terminology Criteria for Adverse Events (CTCAE) version 4 (2010), which was mapped to Medical Dictionary for Regulatory Activities (MedDRA) version 20.0 terms on analysis.

### 2.4. ELISpot Assay (Primary Objective)

Immunogenicity was assessed on cryopreserved PBMC at screening, baseline (i.e., day of first immunization) and weeks 4, 6, 10, 18, and 30 of follow-up by the quantification of T cell responses by an interferon-gamma (IFN-γ) ELISpot assay in a single research laboratory, as described before [10,12]. For patients that restarted ART before week 18, the last PBMC sample off-ART was considered as week 18 for the analysis of the ELISpot results. Briefly, cryopreserved PBMC were stimulated after thawing and overnight (o/n) resting, with consensus B HIV peptide pools in duplicate (15-mers overlapping by 11 aa). The first set of peptide pools matched the HTI immunogen sequence (“IN”); the second set of peptide pools covered the rest of the HIV-1 proteome (“OUT”) [12,16]. Media alone in triplicate was used as the negative control. Stimulations with phytohemagglutinin (PHA-P, final concentration 1 ug/mL, #L-1668, Sigma-Aldrich/Merck KGaA, Darmstadt, Germany) and CEF pools (Cytomegalovirus, Epstein-Barr virus, Influenza virus) final concentration 2 ug/mL, #CTL-CEF-001, ImmunoSpot ^®^, Bonn, Germany) were used as positive controls. Results are expressed as mean number of spot forming cells (SFC)/10^6^ PBMC from duplicate wells. Criteria for analysis were: PBMC viability > 80%, the media control <50 SFC/10^6^ PBMC, positive responses against PHA-P >500 SFC/10^6^ PBMC. ELISpot responses were considered positive in case of >50 SFC/10^6^ PBMC and number of SFC/10^6^ PBMC at least ≥2-fold over media control.

### 2.5. Ultrasensitive Viral Load

A total of 10 mL of EDTA plasma was ultracentrifuged at 170,000× *g* for 1 h at 4 °C. Tubes were equilibrated with 50 mM Tris-Cl, pH 7.6; 150 mM NaCl to a final volume of 12 mL. After centrifugation, 11.2 mL of supernatant was aspirated and discarded. The pellet was thoroughly resuspended in the remaining 800 µL and tested for viral load with the Cobas^®^ HIV-1 test on the Roche Cobas^®^ 4800 system. To account for the concentration of the virus, the obtained result was multiplied by 0.08 (0.8/10).

### 2.6. Viral Reservoir

To explore changes in the HIV reservoir size that could result from activation of HIV-1-specific latently-infected CD4 T cells, total HIV-DNA levels and cell-associated unspliced HIV-RNA (caRNA) [10,12] expression in peripheral CD4 T cells were quantified at baseline and weeks 2, 4, 6, 18, and 30 of follow-up. Both parameters were measured by droplet digital polymerase chain reaction with two different sets of primers (5′LTR and Gag loci) to avoid mismatching, as previously described [17,18]. caRNA was calculated as HIV-RNA copies relative to the housekeeping gene TATA-binding protein (TBP). Total HIV DNA was expressed as log_10_ copies of total HIV DNA per million CD4 T cells measured by the housekeeping gene RPP30.

### 2.7. Statistical Analysis and Interim Analysis

A full overview of the statistical analysis, sample size calculation, and interim analysis is available in Appendix A. Briefly, enrolled participants were included in both an intention-to-treat and safety analysis. Results are displayed as *n* (%) and median, interquartile range (IQR), unless stated otherwise. The primary safety endpoint is described by number and percentage of grade 3 or above AEs. The primary immunogenicity endpoint is described by the total magnitude of HTI specific IFN-γ T cell responses as the sum of SFC/10^6^ input PBMC to all positive HTI peptide pools, and was calculated as log_10_(W6 responses)—log_10_(baseline responses) and log_10_(W18 responses)—log_10_(baseline responses). All analyses were performed two-sided at the 5% significance level, unless otherwise specified. Time until viral rebound, evolution of pVL, and viral reservoir were analyzed using a linear mixed effects model to compare the evolution over the study period. The moderate increase in T cell responses to peptides spanning the HTI sequence at week 8 of the phase I clinical trial [11,12] led to extra vigilance. A protocol amendment in order to perform a futility analysis during the phase IIa trial was implemented. Recruitment was paused after half of the participants were enrolled, followed by an interim analysis on the primary endpoints of the study performed by an independent statistician. In case the standard test statistics for comparing two means was smaller than 2.23, the trial should be stopped for futility. The power of the trial was calculated as 85% in this case.

## 3. Results

Flow chart and design of the study are shown in Figure 1. A total of 38 HIV-infected individuals were screened for eligibility before the interim analysis. From the 35 eligible participants, one withdrew consent, and another was not randomized for logistical reasons; neither were included in the analysis. Thus, 33 participants started and completed the vaccination series (HTI-TriMix *n* = 16; TriMix *n* = 9; WFI *n* = 8). All but one participant were male, with a median age of 42 years, as seen in Table 1. Median time on ART since HIV-1 diagnosis was 6.10 years (IQR 3.35). Two weeks after the last vaccination, 32 participants stopped ART. One participant encountered a SAE after the third vaccination and did not stop ART (see below). Three SAEs were reported and will be discussed later. All participants experienced one or more AEs during the study. There were no significant differences in the total number of participants with AEs between the study arms. Differences in laboratory results AEs were not calculated due to sparse data.

### 3.1. Safety and Tolerability of Intranodal Vaccination

Overall, the procedure and the vaccine were well tolerated. From the total number of 206 AEs (entire study), 18% (*n* = 37) were considered to be definitely related to the procedure or vaccine, but none of them were known grade 3 or above (three of unknown grade), as seen in Table 2 and Appendix A. The most frequently reported AEs (*n* = 32) were short-lasting pain at the site of injection, localized numbness (<48 h) and vasovagal reactions. Twenty-seven percent of all AEs (*n* = 55) were considered to be possibly or probably related to the vaccine or vaccination procedure. This included mild, flu-like symptoms and headache, malaise, chills, and fatigue that lasted for a period <72 h. Four participants had an increase in creatinine kinase (CK), three of them were below grade 3. One of them had a CK increase of >10× upper limit of normal (grade 4) that was considered related to heavy exercise. None of them reported complaints and values normalized without intervention. No other laboratory abnormalities were classified as AE. After the third vaccination, an unrelated SAE was reported for one participant (HTI-TriMix). An acute obstructive cholangitis and associated pancreatitis was diagnosed, and the subject was admitted to a hospital.

During per protocol follow-up and before ATI, pVL was <50 cp/mL, as seen in Appendix A, and no changes in pVL were observed using ultrasensitive viral load measurements, as seen in Appendix A. As shown in Table 1, no changes in CD4 T cell counts were observed throughout the vaccination period.

### 3.2. Safety and Plasma HIV-1 RNA Detection During Treatment Interruption

All 32 participants that underwent ATI had a viral rebound (two consecutive measurements of pVL > 1000 copies/mL separated by at least 15 days, during ATI (one missing measurement for one subject)). First detectable viral load after treatment interruption was observed at a median of two weeks post stop-ART. Hazard ratios of time until viral rebound were not significantly different amongst groups (*p* = 0.18). None of the participants had symptoms suggestive of acute retroviral syndrome during the treatment interruption. Appendix A shows the course of pVL over time during ART after ART resumption. One participant was lost to follow-up after week 10 visit, as he decided not to visit the clinic anymore.

On advice of the treating physician, ART was restarted before week 18 in 20 participants (median 6.2 weeks post ATI, IQR 2.6 weeks; median pVL 5.14 log_10_ cp/mL, IQR 1.68). Ten participants restarted ART per protocol at week 18 (median HIV-RNA 4.80 log_10_ cp/mL IQR 0.58). One participant (WFI) stayed off-ART until week 22, because of a low plasma viremia at week 18 (236 cp/mL) (protocol deviation). At week 22, a SAE was reported for this participant; a worsening of previously known depression and anxiety, which was unlikely to be related to either the vaccine or low grade viremia. Following medical recommendation, ART was reinitiated. None of the participants reached a state of functional cure (pVL < 50cp/mL without ART) at the end of ATI, i.e., at week 18. Based on a locally estimated scatterplot smoothing (loess) graph of log_10_ pVL from treatment interruption until restart of ART, a piecewise linear model was fitted with a breakpoint at 35 days, with nested random effects for subject and site. We found no statistically significant associations for the HTI-TriMix group with pVL (*p* = 0.63).

### 3.3. Safety and pVL Decline During Treatment Resumption

Seven participants (23%) still had detectable pVL at the end of the study (i.e., 12 weeks after treatment resumption). All participants reached pVL < 50cp/mL thereafter, as confirmed by their treating physicians (data not shown, beyond study protocol).

One participant (HTI-TriMix) suffered from a diverticulitis that was complicated by a sigmoidal perforation eight weeks after restarting ART: it was considered a possible immune reconstitution inflammatory syndrome.

### 3.4. Immunogenicity of the Vaccine

At week 6 and compared to baseline, no significant increase in the total frequencies of IFNγ+ IN specific T cell responses in the HTI-TriMix group compared to the WFI group was observed (mean difference −0.07 log_10_ SFC/10^6^ PBMC; 95% CI: −0.35, 0.20; *p* = 0.14, shown for all three groups), as seen in Figure 2. These findings dictated to halt further inclusion of patients for futility. OUT responses behaved similar to IN responses during the vaccination period (week 0–6).

The IFNγ+ T cell responses against the IN peptide showed a trend towards increasing both earlier and to a larger extent in both the HTI-TriMix and TriMix groups as compared to the WFI group. The observed differences did not reach statistical significance. A similar modest increase in the HIV-specific responses against the OUT peptides was observed.

Excluding early restarters, we found no statistically significant differences (*p* = 0.50) in magnitude of HIV-specific T cell responses at baseline and week 18, between HTI-TriMix group and control groups (mean difference compared to WFI 0.07 log_10_ SFC/10^6^ PBMC; 95% CI: −0.51, 0.65).

### 3.5. Changes in HIV Reservoir

We observed no changes in the size of the viral reservoir during vaccination, based on both total HIV DNA and caRNA measurements, as seen in Figure 3. Following ATI, both total HIV-1 DNA and caRNA increased in all study arms. At week 30, i.e., 12 weeks after reinitiating ART, both viral reservoir values were returning to baseline levels. Based on loess smoother graphs of levels of total HIV DNA per 10^6^ CD4 T cells as well as ratios of caRNA versus general transcription factor TBP a piecewise linear model was fitted. For total HIV DNA, this was fitted with breakpoints at 90 and 130. For caRNA a model was fitted with breakpoints at 30, 80, and 150 days. Overall, we did not find a significant effect of HTI-TriMix on the kinetics of the viral reservoir during vaccination, treatment interruption and treatment resumption (Total HIV DNA *p* = 0.99, caRNA *p* = 0.69), compared to the control groups.

## 4. Discussion

The vaccine was shown to be safe and well tolerated as no serious unforeseen safety issues occurred and vaccine related side-effects were short-lived. Unfortunately, after completion of this clinical trial, a coding error was discovered in the plasmid DNA that had been used as template for the mRNA vaccine-construct (see: Appendix B). The plasmid contained an additional and unintended start codon upstream of the HTI open reading frame (ORF). This upstream start codon, present in the mRNA vaccine, allows initiation of an alternative ORF encoding a 15 amino acid peptide. The initiation is favored by the optimal Kozak sequence for this upstream AUG. By contrast, the HTI start codon is not situated in an optimal Kozak consensus. Therefore, this coding error could inhibit or largely reduce initiation and elongation of the downstream HTI ORF [19,20,21]. It is conceivable that the HTI ORF could be suboptimally expressed from the mRNA vaccine.

The vaccine was evaluated preclinically in mice and showed HIV-specific immunogenicity after immunization [11]. Mice were injected with approximately 3 mg of total mRNA/kg bodyweight, while participants of the phase II clinical trial were vaccinated with 13 µg of total mRNA/kg bodyweight. The high dosage in mice may have compensated the poor expression of the vaccine. Of note, safety and moderate immunogenicity of direct injection of naked mRNA have been demonstrated in previous HIV clinical trials, with other immunogens, using dosages which are comparable to the dosage used in humans in the iHIVARNA trial [22,23,24].

The study safety was not compromised, as is reflected by a similar distribution of (S)AEs between study groups. In particular, we observed no individual unforeseen AEs for the HTI-TriMix group during post hoc analysis. Additionally, pVL increased and, concomitantly, CD4 T cell counts declined as expected during a treatment interruption [25]. From the reservoir studies, we observed that the reservoir was approaching baseline values, as observed in previous studies [26]. Even though, in a recent overview of prophylactic and therapeutic DNA and mRNA vaccine studies, it was shown that in some studies, up to 78% of vaccinated subjects experienced adverse events, amongst which 10% were of grade 3. None of the discussed trials that involved mRNA have reached phase 3 research stages yet [27]. Follow-up was given to the notification of the erroneous study product following good clinical practice guidelines, amongst which were notification of participants, competent authorities and institutional review boards.

As recently published, there is consensus that ART interruption is the designated [28,29,30] and feasible [26,31] method to study the effects of a therapeutic intervention (i.e., vaccination) on HIV-1 RNA viral load and reservoir. Currently, no other validated biomarkers are available to study the efficacy of such interventions. This was also detailed in a recent study on virological outcome measures during ART interruption [32]. Some subjects reinitiated ART prior to protocol specifications (i.e., before W18), on decision of the treating physician. It was allowed per protocol for ethical reasons and to prevent avoidable harm to the subjects. However, it was thus not possible to analyze the levels of plasma HIV-1 RNA at a later stage during ART interruption and to compare them to individuals’ pre-ART levels.

Increases in T cell responses were observed after treatment interruption (to both ‘IN’ and ‘OUT’ peptides), probably due to the rebounding virus antigenemia, stimulating the immune system. TriMix may have added to this effect, as we have observed both an earlier and higher increase in immune responses in the TriMix-containing arms versus the WFI arm. This would support the moderate immune responses observed at higher doses in the phase I trial [10,12]. It is not formally studied whether the poor T cell responses to ‘IN’ peptides reflect the result of a lack or low expression of HTI due to the erroneous study product. An additional effect may have come from the presence of RNA molecules in the HTI-TriMix and TriMix control, which nonspecifically stimulates the immune system through triggering of toll-like receptors [33].

Prematurely halting the trial for futility was based on the comparison of ‘IN’ responses from baseline to week 6 using an IFN-γ ELISpot assay during the double-blinded phase of the study. Additional immune assays later confirmed these findings: a CD8+ T cell mediated viral suppressive capacity assay [30] did not show significant changes from baseline, as seen in Appendix A. The poor results kept us from appending the ELISpot with an intracellular cytokine staining, to discriminate CD4 T versus CD8 T cell responses as per protocol.

Still, the combination of subdominant, protective CTL regions, such HTI or other strategies, such as mosaic CTL vaccines aiming at more HIV-1 clades, is supported by several studies [34,35,36].

The erroneous study product made it impossible to draw conclusions on the induction of cellular immune responses against the HTI immunogen. This does not exclude the utility of targeting the more protective regions of the HIV-1 proteome, as more efficient immune responses were observed [37,38]. In vitro and in vivo TriMix studies demonstrated that DCs were more potent and immunogenic than unmodified DC [11,39,40]. Previous studies in HIV-1 viremic progressors showed that levels of CD40L were significantly decreased compared to elite controllers and healthy controls [41]. New insights regarding immune checkpoints might provide auxiliary responses, for instance, the ex-vivo and in-vitro blockade of PD-1, a major regulatory factor for T cell exhaustion in HIV infection, could provide a stronger immune response against the peptides covered in HTI [42]. Last, a recent study has shown distinct differences in subsets of CD4 T cells in those on ART compared to elite controllers and progressors. This might provide other targets for immune therapy and a changed approach to selectively interpret the effects of therapeutic vaccines in future [43].

In conclusion, the HTI-TriMix vaccine was safe and well tolerated. Future studies testing the correct HTI or other conserved HIV immunogens are still warranted and might eventually include additional targets to further enhance the antigen presentation process. An interim analysis for futility and careful participants’ monitoring proved to be of importance to ensure maximal participant safety in our study.

## Figures and Tables

**Figure 1 vaccines-07-00209-f001:**
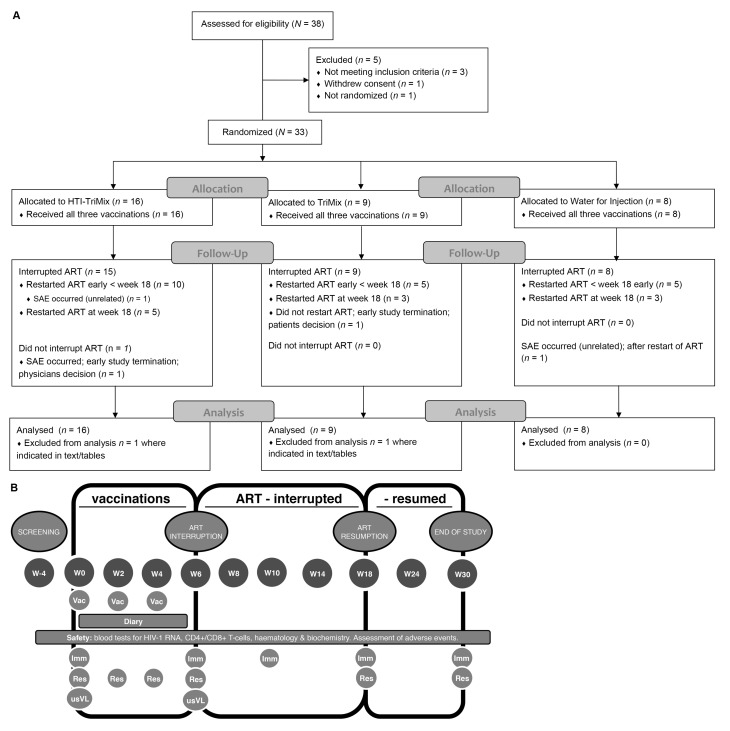
Study flow diagram and study design. (**A**) Consolidated Standards of Reporting Trials (CONSORT) -based diagram [14] delineating the screening and enrollment of study subjects, assignment of study arms, and course of the study. (**B**) Diagram of study procedures. Time points where the assays/study procedures were planned are indicated by: Wx = week x (i.e., W0 = week 0); Vac = vaccination; Imm = Immunogenicity using IFN-γ ELISpot assay; Res = reservoir assessment; usVL; ultrasensitive plasma viral load.

**Figure 2 vaccines-07-00209-f002:**
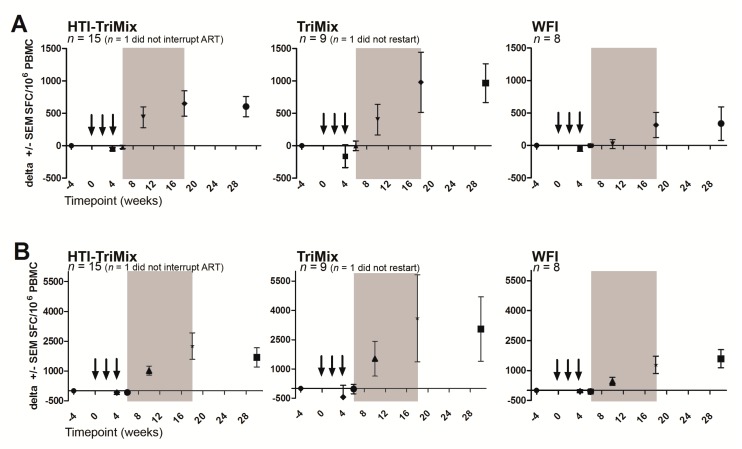
Immunogenicity: IFN-γ ELISpot assay. HIV-specific responses presented as change in spot forming cells (SFC) per 10^6^ PBMC + SEM. Top row (**A**) shows HTI “IN” peptides, bottom row (**B**) shows “OUT” peptides. Arrows indicate vaccinations, grey area indicates ART treatment interruption. Absolute mean (SEM) values at baseline: (**A**): HTI-TriMix 296.7 (137.6); TriMix 430.9 (188.9); WFI 193.5 (99.59) SFC/10^6^ PBMC and (**B**): HTI-TriMix 560.8 (128.3); TriMix 1417 (680.2); WFI 629.0 (252.4) SFC/10^6^ PBMC.

**Figure 3 vaccines-07-00209-f003:**
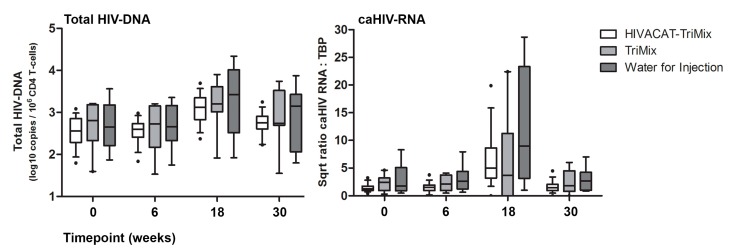
HIV reservoir: total HIV DNA and caRNA. Levels of total HIV DNA and caHIV RNA/TBP. Whiskers show the 10th and 90th percentiles.

**Table 1 vaccines-07-00209-t001:** Clinical characteristics of participants.

		Overall (*N* = 33)	HTI-TriMix (*n* = 16)	TriMix (*n* = 9)	Water for Injection (*n* = 8)	*p*
Median Age (IQR)		42 (17)	44.5 (18)	46.0 (19)	40.0 (19)	NC ^§^
Male *N* (%)		32 (97%)	15 (94%)	9 (100%)	8 (100%)	NC
Supposed Method of HIV Transmission	MSM *	29 (88%)	14 (88%)	8 (89%)	7 (88%)	NC
	Heterosexual	2 (6%)	1 (6%)	0 (0%)	1 (12%)	NC
	IVD ^$^	1 (3%)	0 (0%)	1 (1%)	0 (0%)	NC
	Unknown	1 (3%)	1 (6%)	0 (0%)	0 (0%)	NC
Median HIV-1 pVL Prior to First Start ART (Log_10_ cp/mL) (IQR)		4.65 (0.80)	4.52 (1.05)	4.78 (0.70)	4.93 (1.30)	NC
Median Years from First ART Initiation (IQR)		6.10 (3.41)	4.80 (3.53)	6.93 (5.18)	6.73 (1.16)	NC
Median CD4 Cell Count (IQR) at	^£^ ART Initiation	435 (184)	436 (214)	440 (240)	402 (109)	NC
	Baseline	769 (310)	793 (352)	708 (369)	742 (239)	NC
	Week 6	829 (339)	872 (352)	815 (420)	837 (338)	0.424
	Restart of ART	638 (254)	668 (246)	535 (168)	626 (306)	0.267
	End of Study	846 (323)	908 (411)	748 (314) **	759 (387)	0.313

Characteristics of study participants grouped by treatment arm. Group differences were post-hoc tested using Kruskal–Wallis H test, results presented here are unadjusted for study site and were not corrected for missing data. * MSM: men who have sex with men; ^$^ IVD: intravenous drug use; ^£^ ART: antiretroviral treatment; ^§^ NC: not calculated ** Results of one patient are omitted (did not restart ART, see text and Figure 1).

**Table 2 vaccines-07-00209-t002:** Study safety reported as grade 3 or above adverse events (AEs).

	HTI-TriMix		TriMix		WFI		
	*n*/*N* (%)	95% CI	*n*/*N* (%)	95% CI	*n*/*N* (%)	95% CI	*p*
Local AE (Grade 3 or Above)	0/16 (0.0)	0.0 to 19.4	0/9 (0.0)	0.0 to 29.9	0/8 (0.0)	0.0 to 32.4	NC ^§^
Systemic AE (Grade 3 or Above)	0/16 (0.0)	0.0 to 19.4	1/9 (11.1)	2.0 to 43.5	0/8 (0.0)	0.0 to 32.4	NC
Other Clinical or Lab AE (Grade 3 or Above)	2/16 (12.5) ^☨^	3.5 to 36.0	0/9 (0.0)	0.0 to 29.9	1/8 (12.5)	2.2 to 47.1	0.38

Number of patients (%) who developed grade 3 or above AEs. Because of sparse data, no formal comparison was done for local and systemic AEs. For other clinical or lab AEs, the Cochran–Mantel–Haenszel test was performed, stratified by site, as seen in Appendix A. ☨ one subject with CK increase >10× upper limit of normal. ^§^ NC: not calculated.

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
