# Peer review of "Therapeutic Vaccine in Chronically HIV-1-Infected Patients: A Randomized, Double-Blind, Placebo-Controlled Phase IIa Trial with HTI-TriMixâ€"

_vaccines, 2019, doi:10.3390/vaccines7040209_

Round 1
Reviewer 1 Report
Therapeutic vaccinations against HIV infection merit further investigation. The current manuscript is extension of authors’ previous clinical trial. Based on Preclinical evaluation of an mRNA HIV vaccine, the current manuscript introduced advances on phase IIa trial with HTI-TriMix. Generally, authors completed related clinical trial, analyzed data and drew related conclusions. However, several concerns should be addressed;
1.Cohort design
Why are patients distributed to each cohort regarding to ratio 8:3:3? What about related criteria? Please provide related details or citation.
2.Quality of control
2.1 Whether detection of HT1 expression level was performed in the current trial?
2.2 Besides T lymphocytes, whether expression level of HIV-1 RNA and DNA were detected by PCR in other viral reservoir, such as monocyte or alveolar macrophages, please clarify related details.
3. ELISpot assay
Authors would better provide presentative images for results of ELISpot assay.
Author Response
Reviewer 1 comments for manuscript vaccines-648200:
Reviewer 1
Therapeutic vaccinations against HIV infection merit further investigation. The current manuscript is extension of authors’ previous clinical trial. Based on Preclinical evaluation of an mRNA HIV vaccine, the current manuscript introduced advances on phase IIa trial with HTI-TriMix. Generally, authors completed related clinical trial, analyzed data and drew related conclusions. However, several concerns should be addressed;
Author response:
We want to thank reviewer 1 for reviewing our manuscript and providing us with comments. Please find our answers to the concerns raised in a point-to-point response below.
1.Cohort design
Why are patients distributed to each cohort regarding to ratio 8:3:3? What about related criteria? Please provide related details or citation.
Author response:
in phase IIa exploratory studies it is preferred to have more patients included in the active arm than in the control arms. For our studies we had to take into account that there were 2 control groups without potential efficacy (both TriMix and WFI). Therefore it was unethical to give to 2/3 parts of patients a “placebo”. The distribution like 8:3:3, permitted that less than half of the patients received placebo and the bigger half (57%) the drug under investigation. This was followed by a sample size calculation (which was repeated for the interim analysis) as detailed in the manuscript. We added an extra line in section 2.2, that refers to 2.7 and Supplementary data 2.
2.Quality of control
2.1 Whether detection of HT1 expression level was performed in the current trial?
Author response: HTI-TriMix expression was evaluated in preclinical work of Guardo et al. (see introduction in manuscript and response to editor). In the phase I and II trials we assessed HTI specific responses using ELISpot assay peptide pools as is detailed in 2.4, referred to as “IN” peptides. These were compared to responses to “OUT” peptides (i.e. not included in the HTI construct). The data are summarized in figure 2; also refer to one of the next comments.
2.2 Besides T lymphocytes, whether expression level of HIV-1 RNA and DNA were detected by PCR in other viral reservoir, such as monocyte or alveolar macrophages, please clarify related details.
Author response: we studied the reservoir in peripheral blood CD4 T-cells, as these are considered the major (though, not unique) peripheral reservoir. If we couldn’t detect changes in this reservoir, it would be unlikely that the reservoir will be altered in other cell types in the peripheral blood. Monocytes are really hard to be infected with HIV. Their number in peripheral blood will be much lower than CD4 T cells, which thus reaches the technical limits of detection. In our hands, peripheral monocytes had no detectable HIV DNA (unpublished data). Adequate study of alveolar macrophages would only have been possible by performing multiple bronchoscopies (i.e. baseline vs follow-up). Though this would be interesting, proposing this intensive procedure would have led to ethical issues and potential subjects not wanting to participate.
3. ELISpot assay
Authors would better provide presentative images for results of ELISpot assay.
Author response:
Our figure 2 with ELIspot results displays the outcome of the primary endpoint for immunogenicity, which was defined as change in spot forming cells (SFC) compared to baseline, for the individual groups. This clearly shows that the vaccination did not induced extra immunogenicity. Only after treatment interruption (ATI) the number of HIV-specific SFC increased both to IN and OUT peptides.
We tried other ways displaying this result, but these were not more clear in making this point.
For clarity we propose to make the ATI shaded, to emphasize this important phase more clearly.
We corrected a minor error in text.
We added absolute baseline values for the ELIspot.
Reviewer 2 Report
This is a nicely written report on negative results of a therapeutic vaccine administered via intra nodal injection in chronically infected HIV individuals. However, the fact that the manufacturing error wasn't noticed until after the completion of phase I and II trials does raise concerns. Presumably, sequencing of the vaccine antigen should be part of the product release tests and procedures and any discrepancies should have been discussed with regulatory agencies prior to the start of these clinical trials. Can the authors please comment on any interactions with the regulators around the erroneous product?
It is a shame that the problems with the study vaccine make your immunology results uninterpretable. Nonetheless, the manuscript provides valuable information on safety and feasibility of intra-nodal vaccine administration.
The authors state that HTI expression has not been assessed, and I wonder if you could assess it before attributing the lack of T-cell responses to the erroneous product.
Line 67: typographical error where 'experiments' is written twice.
Table 1 says that group differences were tested using Kruskal-Wallis H test, but the results of the statistical testing are not shown in the table or commented on the footer. There is also a typo on 'Krusal'.
Table 2. Presumably the grade 3 'other clinical or lab AEs' were considered at least possibly related to the IMP or study interventions. Can you clarify if this is the case, and also clarify if the 3 events reported in the table are the same as the 3 participants with an increase in CK?
The study diagram implies participants would necessarily resume ART at week 18. However the authors state that 1 participant stayed off ART until week 24. Was this a protocol deviation? Can you expand on the rationale for not reintroducing ART at week 18 as originally proposed? Could the prolonged suspension of ART had been the trigger for worsening of his anxiety/depression episode?
Finally, the fact that 20 out of 33 randomised participants had their ART resumed before the proposed timepoint on medical grounds contradicts the argument that ART interruption is the most feasible method for assessing the effects of therapeutic interventions and it might be worth further discussing the practicalities of ART interruption.
Author Response
Reviewer 2 comments for manuscript vaccines-648200:
This is a nicely written report on negative results of a therapeutic vaccine administered via intra nodal injection in chronically infected HIV individuals. However, the fact that the manufacturing error wasn't noticed until after the completion of phase I and II trials does raise concerns. Presumably, sequencing of the vaccine antigen should be part of the product release tests and procedures and any discrepancies should have been discussed with regulatory agencies prior to the start of these clinical trials. Can the authors please comment on any interactions with the regulators around the erroneous product?
AUTHOR RESPONSE:
We want to thank reviewer 2 for reviewing our manuscript and providing us with comments.
We share the concerns. The consortium was informed of the erroneous study product by the manufacturer only after the phase I and II trials were completed, in November 2018. We immediately informed the authorities and asked for their demands. Upon their request we consulted the Data Safety Monitoring Board and a panel of experts working in field of HIV research, immunology, clinicians, and industry for their advice. In summary, no adverse events were related to the erroneous study product and it was discouraged to further study HTI expression in vitro as that would not provide definite answers. We have point-to-point addressed all questions of the competent authorities and have shared study data in the European Union clinical trial register (EudraCT), as well as Clinicaltrials.gov. A footnote was included in all scientific communication, as was requested by the authorities.
Thus we have acted prompt and correctly to the demands of the supervising committees and authorities. It should be noted that when we prepared the phase II trial, all product information and controls as gathered in the IMPD and Investigator’s Brochure had already been submitted and approved for the phase I trial. The manufacturer of the study product had all required GLP/GPP licenses. The manufacturer was visited by a contract research organization (CRO) to fulfill GCP compliance prior to study start (on behalf of the sponsor). The product was sequenced before release, but the sequence was only compared to the template sequence of a previous batch, that contained the same error. The manufacturer has installed revised procedures, to avoid future mistakes.
It is a shame that the problems with the study vaccine make your immunology results uninterpretable. Nonetheless, the manuscript provides valuable information on safety and feasibility of intra-nodal vaccine administration.
AUTHOR RESPONSE:
Thank you for this comment, and indeed we would like to share the data with the (scientific) community, amongst which is safety and feasibility of the vaccine administration.
The authors state that HTI expression has not been assessed, and I wonder if you could assess it before attributing the lack of T-cell responses to the erroneous product.
AUTHOR RESPONSE:
We could not confirm HTI expression in vitro, using Western blots. We interpreted this as a lack of specificity of the antibodies that we used. However, HTI-TriMix expression was determined in the preclinical work of Guardo et al. (see introduction), by positive ELIspot results in HTI naïve mice. The reviewer is right in that we cannot formally attribute the lack of T-cell responses to the lack of or low expression of the erroneous product. We have adjusted the wording in the text. However, we do not have the study product available for further testing.
Line 67: typographical error where 'experiments' is written twice.
AUTHOR RESPONSE: corrected
Table 1 says that group differences were tested using Kruskal-Wallis H test, but the results of the statistical testing are not shown in the table or commented on the footer. There is also a typo on 'Krusal'.
AUTHOR RESPONSE:
please excuse us for the sloppiness. Typo was corrected and correct data is now in the table.
Table 2. Presumably the grade 3 'other clinical or lab AEs' were considered at least possibly related to the IMP or study interventions. Can you clarify if this is the case, and also clarify if the 3 events reported in the table are the same as the 3 participants with an increase in CK?
AUTHOR RESPONSE:
We reported the CK increase in the main body text as this was outstanding from the other laboratory results. The other laboratory results showed no relevant findings. Only one participant with an increase in CK is reported in the table (grade 4). Others included a penile ulcer, trigeminal neuralgia, anxiety, gall stones (later reported as SAE) (see Supplementary data). As these other grade 3 or above were considered unrelated to the IMP or interventions, we did not discuss them in the text. For clarification however, we added the classification of the CK increase in the body text and indicated it in the table.
The study diagram implies participants would necessarily resume ART at week 18. However the authors state that 1 participant stayed off ART until week 24. Was this a protocol deviation? Can you expand on the rationale for not reintroducing ART at week 18 as originally proposed? Could the prolonged suspension of ART had been the trigger for worsening of his anxiety/depression episode?
AUTHOR RESPONSE:
The protocol specified restart of ART in case HIV-1 plasma viral load > 50 copies/mL on W18 or later (i.e. W18-W30). Since this was indeed a protocol deviation, we had to report it in the manuscript. We now added “protocol deviation” to the text, for clarification. As it was only a relative low level viremia (compared to other observations, i.e. pVL 5.14 log10 cp/mL, IQR 1.68), we consider it unlikely to be related to the viremia. In addition, no other complaints such as flu like symptoms etc. or other relevant complaints were reported.
Finally, the fact that 20 out of 33 randomised participants had their ART resumed before the proposed timepoint on medical grounds contradicts the argument that ART interruption is the most feasible method for assessing the effects of therapeutic interventions and it might be worth further discussing the practicalities of ART interruption.
AUTHOR RESPONSE:
Indeed, early restart of ART in our study might contradict ART interruption as “most feasible” method. However, from recent consensus meetings (see reference in text; Julg et al. Lancet HIV 2019 vol 6 (4), e259-e268) it was agreed that validated biomarkers predictive of virological control once ART is stopped are not yet available. ART interruption is the only way to test the efficacy of new therapeutic interventions, and it was concluded that this strategy is not yet irreplaceable at inducing HIV suppression in the absence of ART. Another discussion is about which is the "gold standard" virological outcome measure. So far there is no general consensus as different studies use different virological end-points such as time to rebound, set point, peak viral load. In text we now have elaborated more on this possible limitation and added a citation to a recent paper (Feher et al. Open Forum Inf Dis 2019; https://doi.org/10.1093/ofid/ofz485 ) on evaluating different virological outcome measures during ATI. This should provide better context to the reader.
Round 2
Reviewer 1 Report
Authors answered all concerns, and improved the quality of manuscript.